# RePlant Alfa: Integrating Google Earth Engine and R Coding to Support the Identification of Priority Areas for Ecological Restoration

Narkis S. Morales [1,*], Ignacio C. Fernández [2], Leonardo P. Durán [3,4] and Waldo A. Pérez-Martínez [3,4]

1   Departamento de Ecosistemas y Medio Ambiente, Facultad de Agronomía e Ingeniería Forestal, Pontificia Universidad Católica de Chile, Santiago 7820436, Chile
2   Centro de Modelación y Monitoreo de Ecosistemas, Facultad de Ciencias, Ingeniería y Tecnología, Universidad Mayor, Santiago 7500994, Chile
3   Escuela de Ingeniería Forestal, Facultad de Ciencias, Ingeniería y Tecnología, Universidad Mayor, Santiago 7500994, Chile
4   Hémera Centro de Observación de la Tierra, Facultad de Ciencias, Ingeniería y Tecnología, Universidad Mayor, Santiago 7500994, Chile
*   Correspondence: nsmorale@uc.cl

**Abstract:** Land degradation and climate change are among the main threats to the sustainability of ecosystems worldwide. As a result, the restoration of degraded landscapes is essential to maintaining the functionality of ecosystems, especially those with greater social, economic, and environmental vulnerability. Nevertheless, policymakers are frequently challenged by deciding where to prioritize restoration actions, which usually includes dealing with multiple and complex needs under an always limited budget. If these decisions are not taken based on proper data and processes, restoration implementation can easily fail. In order to help decision-makers take informed decisions on where to implement restoration activities, we have developed a semiautomatic geospatial platform to prioritize areas for restoration activities based on ecological, social, and economic variables. This platform takes advantage of the potential to integrate R coding, Google Earth Engine cloud computing, and GIS visualization services to generate an interactive geospatial decision-maker tool for restoration. Here, we present a prototype version called "RePlant alpha", which was tested with data from the Central Zone of Chile. This exercise proved that integrating R and GEE was feasible, and that the analysis with at least six indicators for a specific region was also feasible to implement even from a personal computer. Therefore, the use of a virtual machine in the cloud with a large number of indicators over large areas is both possible and practical.

**Keywords:** google earth engine; R coding; GIS; restoration; decision-making

## 1. Introduction

Human activities have had a profound impact on the earth's ecosystems on a global scale. This impact has been so great that several authors have argued that we are in a new geological epoch: the Anthropocene [1]. The historical pressure on ecosystems has been increasing hand in hand with population growth, mainly driven by the demand for resources to cover essential needs such as food and shelter [2,3]. These have led land transformation and soil degradation to be among the greatest threats to the resilience of ecosystems worldwide [4]. Thus, developing strategies to help restore degraded ecosystems is an urgent task to help keep the planet in healthy ecological conditions to support biodiversity, tackle climate change, and promote human development [5]. Additionally, the United Nations has made an explicit call to implement restoration activities, naming the period 2021–2030 the "UN Decade on Ecosystem Restoration" (). Nevertheless, developing restoration activities will be challenging, particularly when they need to be interwoven with a myriad of other uses in socio-ecological landscape mosaics [6]. Under these circumstances, it is

key to identify areas that are likely to maximize the benefit provided by restoration and minimize the social and economic costs of implementation [7]. Hence, developing tools to help overcome the challenges of implementing restoration activities in socio-ecological landscapes is key to increasing the effectiveness of a country's specific actions.

One of the main challenges to identifying the best areas to implement restoration activities is the large number of variables that need to be jointly assessed, which usually include ecological, economic, social, and institutional factors [8]. Spatial prioritizations like this could be addressed with the use of multi-criteria spatial decision analyses, which integrate different geospatial variables through a system of interaction rules that combine them to generate information for decision-making [9–11]. Such an approach has been recently proposed to identify global priority areas for restoration [12] and has also been used for prioritizing areas for restoration at finer scales (e.g., [13–15]).

However, a pervasive characteristic common to many methods for environmental decision-making is the inadequate inclusion of stakeholders in the decision-making process [16]. This is also a prevalent issue in the prioritization of restoration projects, as most published studies do not adequately include stakeholders [17]. While this problem could be solved by investing time and resources to ensure the participation of an inclusive and representative group of stakeholders, the prioritization outcomes would only be valid for the specific context (e.g., geographical, ecological, economic, social, and institutional) in which the decision-making process took place. Therefore, researchers could be tempted to reduce the participation of stakeholders in spatial prioritization methods to make the process simpler and produce more generalizable outcomes, decreasing the legitimacy of the results [16]. This may be particularly true for spatial analyses, as the tasks of data collection, processing, and analysis are intensive in terms of time and computational resources. Due to the complexity of the different regional scenarios, it is tempting to skip the regional differences and focus on the generalities, which can have negative impacts on the restoration process (e.g., [8,18,19]). For example, Chile is a long and narrow country that has a north-south and west-east climatological gradient, a large diversity of vegetational communities, a large urban concentration, and sectors dominated by rural populations. It also has contrasting drivers of land degradation and productive activities throughout the territory. While in some regions mining is the principal economic activity, in others forestry and agriculture are dominant [20]. If we only include the mining industry in the environmental decision-making process, results will only be valid for regions where that activity is dominant.

However, the increasing availability of free spatial information, geospatial processing packages, and cloud computing capabilities has expanded the frontiers of what can be done in environmental spatial analysis [21,22]. These new tools may offer an opportunity to generate a new era for multi-criteria spatial decision analysis, where researchers focus on developing on-line interactive decision-making platforms, and stakeholders operate them in real-time to evaluate different scenarios based on specific context. In this work, we present the preliminary results of an ongoing project aimed at developing a free online interactive geospatial decision-making platform for guiding the selection of areas for ecological restoration in Chile. The prototype, called "RePlant alpha" uses R coding for requesting and managing the data, and for framing and running the decision-making algorithms, Google Earth Engine (GEE) for collecting and preprocessing satellite-based spatial data, and an online viewer for interactively showing the results.

## 2. RePlant alpha

The current prototype is a computer-based platform that integrates the flexibility of R coding with the cloud computing capabilities of Google Earth Engine (Figure 1). In general terms, the platform works based on a set of predefined indicators (including the ecological, social, and economic dimensions), which, depending on their characteristics, are processed on GEE or on a local computer. Currently, all satellite-based indicators are cloud-computed in GEE, whereas the other indicators are computed from geospatial data stored on a local

computer. Once all the indicators have been computed and standardized into compatible units, they are integrated through a spatial multi-criteria analysis (SMCA). During the integration process, a sensitivity analysis is performed to help define the weights used in the SMCA. In addition, an optional routine is available to run all the possible combinations of weights as a pre-cached database of results or to build a metamodel. Results are then uploaded to a web-based mapping platform (ArcGis Online) to produce interactive maps (Figure 1).

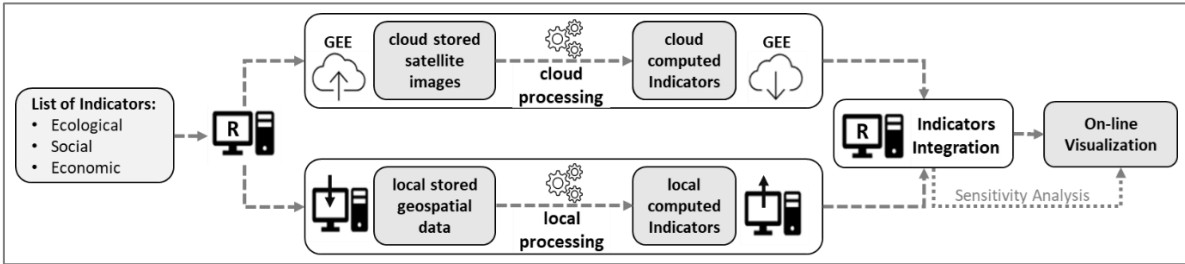

**Figure 1.** Diagram representing the main processes implicated in generating the prioritization of areas for restoration in the RePlant alpha platform.

## 3. RePlant alpha in Action

With the aim of showing the potential of the current prototype, we used RePlant alpha to identify areas for restoration within the administrative region of Valparaíso, in Central Chile (Figure 2). This region of 16,396 km$^2$ has a Mediterranean climate, with annual rainfalls of 350 mm concentrated predominantly during the winter months. The original vegetation is mainly represented by schlerophyll forests and shrublands, which are currently relegated to higher and steeper areas, while lower and flat areas have been largely transformed for urbanization, agriculture, and forestry. The changes in landscape composition and structure have made this region prone to forest fires, which have intensified the negative impacts of land transformation and degradation.

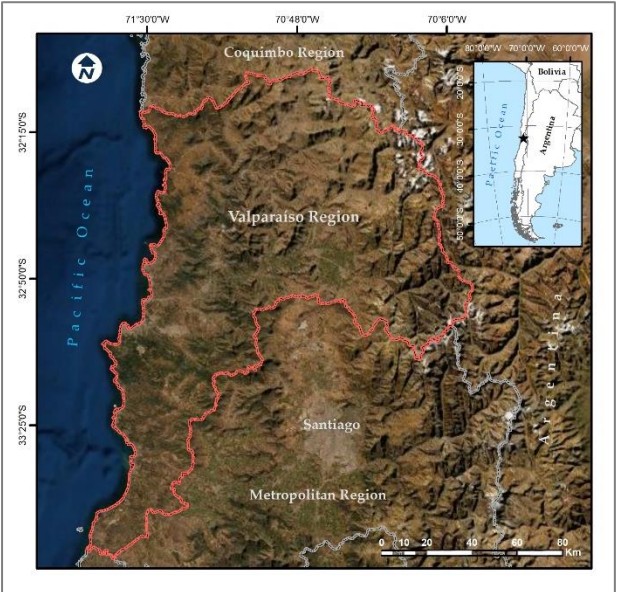

**Figure 2.** Map showing the area where the prioritization was performed. The Valparaiso Region is shown with a red border.

### 3.1. Selection of Indicators

We selected seven indicators to test the platform. We decided to use indicators that could be computed with freely accessible data but were at the same time useful to determine areas with restoration needs. In order to select these indicators, we considered not only

factors involved in the potential success of restoration processes, but also those related to the social impact and operational feasibility of their implementation in the area [15,23]. The indicators were divided into three main categories: ecological, social, and economic (Table 1). The selected indicators were: normalized difference vegetation index (NDVI), differenced Normalized Burn Ratio (dNBR), slope and potential erosion indexes, population density, and distance to roads.

**Table 1.** List of selected main indicators including classification and basic information.

| Indicator | Category | Description/Objective | Source | Scale |
|---|---|---|---|---|
| NDVI | Ecological | Calculation of NDVI over the last 5 years for each sample unit to evaluate the trajectory of vegetation in the sector | Landsat | 30 m/pixel |
| dNBR | Ecological | Calculation of dNBR based on the difference between pre- and post-fire seasons, to estimate damage severity | Landsat | 30 m/pixel |
| Slope | Ecological | Calculation of slope in percentage to estimate erosion potential | Digital Elevation Model/Aster | 30 m/pixel |
| Potential erosion index | Ecological | Estimation of potential erosion are based on an empirical qualitative model (IREPOT) and represent risk of erosion using specific characteristics of the studied areas. | Spatial data/IDE | 1/50,000 |
| Population Density | Social | Estimation of population density to calculate the potential for community support | Spatialized census data/IDE | Census block |
| Proximity to roads | Economic | Generation of distance map to main and secondary roads to estimate ease of access to the area | Spatial data/IDE | 1/10,000 |

The NDVI and dNBR were chosen to assess vegetation changes (during the year) and detect burned areas (December to March). We chose to use slope because steep areas should have priority over flat ones due to the higher risk of future soil loss, landslides, and floods. However, due to the fact that not all the areas with high slopes have the same characteristics of erodibility and rainfall erosivity, we added the potential erosion index to account for that variability in steep areas. Since local community support for any restoration project is important [8], population density could give an estimate of the potential community participation in any given area. Finally, the proximity to roads indicator was selected to give greater value to sites closer to roads and prioritize areas that have fewer logistical limitations to carry supplies, both in terms of vegetative plant material and human resources.

*3.2. Indicator's Building Process*

3.2.1. Cloud Data Processing

The main objective of this proof of concept was to automate the compilation and analysis of the information needed to build some of the indicators. The GEE can be accessed through a web-based integrated development environment (IDE) or by a Python application programming interface (API). In order to integrate GEE with R, we use the package "Reticulate," which provides an interoperability layer to use Python in a R session [24]. In addition, the GEE Python API was installed using the package "rgee" [25]. These two packages allowed us to connect to GEE with our credentials using Python in an R session.

A shapefile of the study area was uploaded to GEE to delineate the area of interest. The Tier 1 Landsat 8 image collection was selected and later processed to eliminate unwanted pixels (e.g., clouds, shadows) using a mask based on the quality band (pixel_qa). The preprocessed images were used to calculate monthly NDVI and NBR during 2017.

The monthly indexes were then compiled into one composite raster, representing the median of 12 months for each index. The slope was derived from a digital elevation model available in the GEE datasets (SRTM Plus). The resulting composites were cut to the extension of the study area and then exported as a raster to the local R session.

### 3.2.2. Local Data Processing

The methodology described in this section uses the "sp" [26], "sf" [27], "stars" [28], "tidyr" [29], and "Raster" packages in R [30].

The data needed to build the rest of the indicators was not available in GEE. Therefore, they were built using a local working directory and produced using R GIS tools. Some indicators, such as population density (PD), distance to roads (DR), and potential erosion index (PEI), were in a vectorial format; therefore, we transformed them to a raster format.

The PD was derived from a comm- delimited file that was then filtered, leaving only the continental territory (i.e., removing outer islands) as part of the study. The resulting data was then spatialized, assigning the proper data to each administrative commune using the "stars" package [28], and transformed into a raster file. The DR was derived from a vectorial layer that included all the road infrastructure of the study area. The vectorial layer was rasterized by calculating the distance from cells that were not identified as populated areas to cells identified as roads, using the distance command from the package "raster" [30]. In addition, the PEI corresponded to the results delivered by a potential erosion risk index model (IREPOT, by the acronym in Spanish) [31]. The model also integrates different variables, including topographic-hydrological, rainfall aggressiveness, soil, and vegetation. All the variables were later integrated into a single qualitative index, which was represented by four qualitative categories of potential risk (none or low, moderate, severe, and very severe). Each category was associated with a risk class ranging from 1 to 4 (low to very severe). This final layer was rasterized using the risk class.

### 3.3. Integration of Indicators

All generated raster layers were checked, reprojected, and resampled if needed. The new raster layers were then normalized so that they were in the same range of values between 0 and 1. This was done using the min-max rescaling method. Once all the layers were on the same numerical scale, they were integrated in a SMCA using different weights. Due to the fact that the use of weights can be subjective, we included a sensitivity analysis. This analysis can give the user some insight regarding the influence of each weight on the final SMCA results. The weight values were changed one at a time by ±0.2 starting from 0 to ±1. However, for each change in weight, a raster was built that was later compared to a reference raster map representing a full model including all the indicators without weights. To represent all the comparisons, a contingency table showing the agreement (%) for each indicator was made to summarize the results using the package "diffeR" [32] (Table 2).

**Table 2.** Agreement (%) results for changes in weights of each indicator (ranging from −1 to 1). The analysis corresponds to a comparison from a raster representing a weight change for a specific indicator and the raster representing the baseline model.

| | Weights | | | | | | | | | | |
|---|---|---|---|---|---|---|---|---|---|---|---|
| **Indicators** | −1 | −0.8 | −0.6 | −0.4 | −0.2 | 0 | 0.2 | 0.4 | 0.6 | 0.8 | 1 |
| Population Density | 98 | 98 | 98 | 99 | 99 | 99 | 99 | 100 | 100 | 100 | 100 |
| EJH | 4 | 9 | 16 | 25 | 34 | 45 | 57 | 66 | 75 | 86 | 100 |
| Erosion | 40 | 41 | 43 | 46 | 53 | 62 | 72 | 79 | 84 | 90 | 100 |
| NBR | 16 | 26 | 36 | 88 | 45 | 54 | 63 | 70 | 79 | 88 | 100 |
| NDVI | 0 | 2 | 7 | 17 | 30 | 42 | 53 | 63 | 73 | 85 | 100 |
| Proximity to roads | 2 | 2 | 3 | 3 | 4 | 12 | 32 | 50 | 66 | 80 | 100 |
| Slope | 0 | 1 | 4 | 10 | 17 | 25 | 40 | 55 | 68 | 82 | 100 |

Following the results from the sensitivity analysis, the weights for each indicator were defined according to their effect on the model. For example, for indicators that had a low

impact (e.g., population density), low weight values were assigned (0.1). In contrast, for indicators with a high impact on the outcome (e.g., proximity to roads) and slope, low weight values were assigned. The rest of the indicators were set to 0.2. The prioritization raster or priority index from the SMCA was reclassified in quartiles for ease of interpretation and vectorized to make it compatible with a vectorial territorial administrative layer.

### 3.4. Cache of Different Weights Combinations

The code includes an option where the SMCA results for each possible combination of weights can be cached on the computer, in the cloud, or on a server. However, we do not recommend its use because the number of weight combinations is really high (approximately 44 million different combinations for 7 indicators). In our case, the resulting rasters from each weight combination had a size of 7 MB, which multiplied by the number of different weight combinations would require 308 TB of storage.

### 3.5. Visualization Platform

The vectorial layers representing the prioritization and the territorial administrative divisions were manually uploaded to web-based mapping software (Arcgis Online®, ESRI). The platform has three interactive layers that the user can explore. A main layer representing the areas with a high prioritization index (>0.6) called priority areas; a secondary layer showing the total priority area per commune called priority rate per commune (%); and a third layer that corresponds to the reclassified prioritization index. Each layer has a caption with its respective index or percentage value (Figure 3).

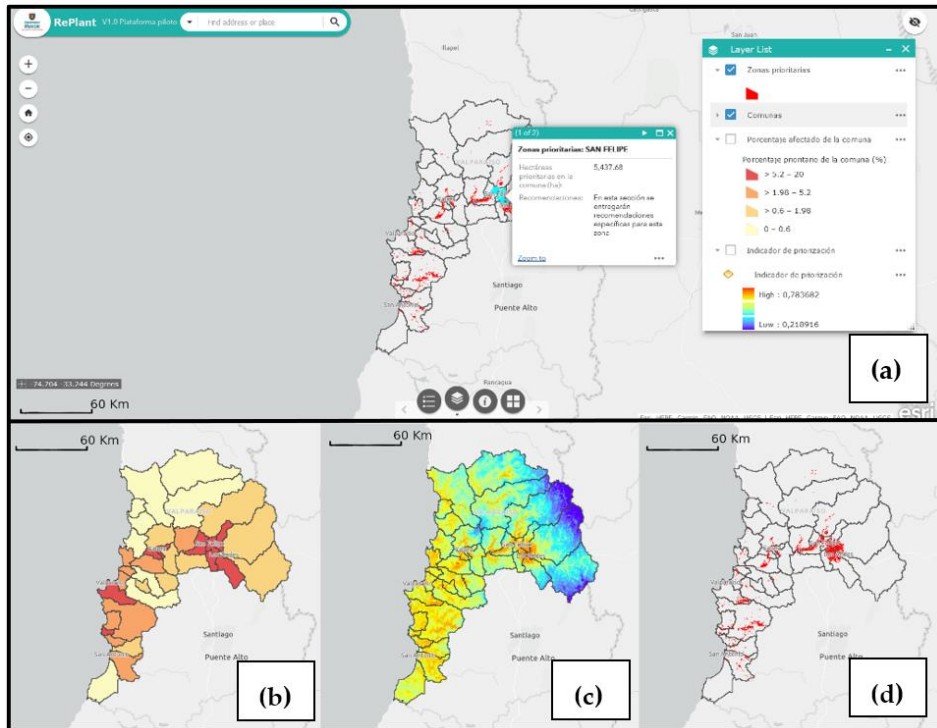

**Figure 3.** General view of the "Replant alpha" platform and the priorization results presented in the ArcGIS web map. (**a**) The platform where the results to be displayed can be manipulated, (**b**) Priority percentage of the commune (%), (**c**) prioritization index (0 to 1) and (**d**) priority restoration zones representing areas with high restoration priority.

## 4. Results

### 4.1. SMCA Output

The prioritized areas corresponded to 18% of the total area of the Valparaiso region (total region area 160.6977 ha, not including insular areas). The provinces with a higher

level of degradation were Valparaiso and San Antonio, with 5.5 and 4% of the total region area, respectively (Appendix A). Coincidentally, the top three communes with the highest degraded area were located in these provinces. Valparaiso province has two communes, Casablanca (2.9%) and Valparaiso (1.1%), with the highest values of degradation. San Antonio Province had one commune (San Antonio) with 1.3 % of the total area of the region in need of restoration. The rest of the communes in the region had less than 1% of the total area of the region classified as being in need of restoration (Appendix A). It is important to note that even though the proportional contribution of the different communes to the total could be seen as small in comparison with the total area of the region, some communes have more than 40% of the area to restore per province. All 35 communes are in need of restoration, although their contributions to each degraded province can vary from 2.3 to 53.1%.

*4.2. Processing Times*

On a top-of-the-line computer, the whole script for seven indicators takes approximately 40 min (not including permutations of the different weights or caches of different combinations). However, it is important to consider that running the whole script will not always be necessary, as many of the processed indicators could be stored as cached data; thus, requiring to run the whole script once a year or when updated information is available (e.g., new satellite images, updated social data). In any case, if more processing power is needed, it can be scaled to a virtual machine on any cloud service without issue.

*4.3. Sensitivity Analysis*

The sensitivity analysis showed that population density is the less sensitive indicator, meaning that changes in the weight associated with this indicator have a minimal impact on the prioritization model. In contrast, the most sensitive indicators, proximity to roads and slope, when not considered in the prioritization model (weight equal to zero), had significant impacts on the model with only a 12% and 25% agreement, respectively. In any case, most of the indicators were quite robust to weight changes (0.8 point reduction) (Table 2).

**5. Future Improvements**

*5.1. Code and Indicators*

The code was written with only seven indicators, as this was a proof of concept. Including more indicators can vary from a straightforward process to requiring major coding. In order to avoid this issue, all the code that includes absolute paths and filenames must be changed to relative paths. This change will simplify the directory structure necessary for the different data files and allow the code to run on any device or cloud setup. Relative indicator names are needed to make the code shorter and faster, especially for functions in a loop and processor intensive tasks. Finally, these changes will help the future platform administrator make changes or updates easily. While the current prioritization module has only seven basic indicators, a newer version of the platform is being developed to add 11 new indicators to the SMCA (Table 3). The future versions of the platform will allow the users to select the indicators and weights used in the SMCA that are more appropriate to their needs.

**Table 3.** General information about the indicators that a future version of the platform will include.

| Indicador | Category | Focus | Description/Indicator Objective | Data Source | Data Source Scale |
|---|---|---|---|---|---|
| NDVI temporal changes | Ecological | Land degradation | NDVI was calculated using the last 5 years for each sample unit to evaluate the trajectory of vegetation in the sector | Landsat | 30 m/pixel |
| dNBR | Ecological | Land degradation | Calculated using the difference between pre- and post-fire seasons, to estimate damage severity | Landsat | 30 m/pixel |
| Land use | Ecological | Land degradation | Classification of land uses (e.g., urban, agricultural, forestry, natural) according to their potential for restoration | National Native Forest Inventory | 1/5000 |
| Slope | Ecological | Abiotic factors (erosion proxy) | Calculation of slope in percentage to estimate erosion potential | Aster Digital elevation model | 30 m/pixel |
| Aspect | Ecological | Abiotic Factors (HR and T° proxy) | Calculation of exposure in degrees to estimate potential soil moisture/temperature conditions. | Aster Digital elevation model | 30 m/pixel |
| Proximity to Priority Conservation Sites | Ecological | Landscape Continuity | Generation of a map including proximity to a priority site and/or areas part of priority sites to prioritize landscape continuity | Chilean Geospatial Data Infrastructure | 1/10,000 |
| Proximity to SNASPE | Ecological | Landscape Continuity | Creation of a map of belonging and proximity to priority protected areas of the State to prioritize landscape continuity | Chilean Geospatial Data Infrastructure | 1/10,000 |
| Proximity to a native vegetation fragment | Ecological | Landscape Continuity | Map representing the distance to native vegetation patches to prioritize landscape continuity | National Native Forest Inventory | 1/5000 |
| Vegetation cover type | Ecological | Landscape Diversity | Vegetation cover map categorized by functional types (herbaceous, evergreen, deciduous), to prioritize functional groups to be reforested | Landsat | 30 m/pixel |
| Particulate Matter | Social/Economic | Social Impact | Urban particulate matter maps for 2.5- and 10-micron, o establish priority zones for restoration | National Air Quality Information System and proprietary network of contamination monitor (Fernández IC) | 1/10,000 |
| Land tenure | Social/Economic | Social Impact | Map of properties and/or land ownership to evaluate accessibility to areas with restoration priority | Chilean Natural Resources Information Center | 100 m/pixel |
| Multidimensional Poverty | Social | Social Impact | District map of multidimensional poverty to prioritize by level of socioeconomic vulnerability | National Socioeconomic Characterization Survey | Census block |
| Extreme Poverty by Income | Social | Social Impact | District map of extreme poverty to prioritize by level of socioeconomic vulnerability | National Socioeconomic Characterization Survey | Commune |
| Unemployment Rate | Social | Social Impact | District unemployment map to estimate the need for jobs in affected areas | National Socioeconomic Characterization Survey | Commune |

**Table 3.** *Cont.*

| Indicador | Category | Focus | Description/Indicator Objective | Data Source | Data Source Scale |
|---|---|---|---|---|---|
| Percentage of population with Higher Education | Economic | Human Capital | District unemployment map to estimate the need for jobs in affected areas | National Socioeconomic Characterization Survey | Commune |
| Distance from population centers | Economic | Logistic | Map representing the distance to population centers to estimate the ease of obtaining inputs | Chilean Geospatial Data Infrastructure | Commune |
| Population Density | Economic | Logistic | Estimation of population density to estimate the potential to recruit local labor | Chilean Geospatial Data Infrastructure | Commune |
| Proximity to roads | Economic | Logistic | Map of distances to main and secondary roads to estimate ease of access to the area | Chilean Geospatial Data Infrastructure | 1/10,000 |

*5.2. Data Visualization*

In terms of the data visualization, currently the results are not transferred automatically. The approach to use will depend on who hosts the final version of the platform. There are three options on how to transfer the data to a visualization module. One option is to build a tailored platform and graphic interface using a hybrid composition of web development, databases, and cloud computing. For example, a web service using HTML5, CSS3, Javascript, JQuery, and Python3 technologies, through the Django Framework, for the presentation layer (frontend) and the processing layer (backend), would work well. In data storage and query response, a MySQL database can be used, while the complete solution could be built on a cloud service such as Google Cloud or Amazon Web Services (AWS), including Google Compute Engine (GCE) instances and Cloud SQL services. This solution is more cost-effective because the only fixed cost in the long term would be the use of cloud services. Another option is to keep using the ArcGis web-based mapping solution (ArcGis online ®) together with R-ArcGIS Bridge and ESRI's Web AppBuilder framework to provide a web-based frontend. A similar solution would be the use of RStudio and packages of different cloud services, such as Shiny, Shinyapps, or RStudio Connect. Although these are the best solutions in terms of integration, they require a subscription to the services, and the costs can escalate quickly depending on the services demanded by users and the number of users connected.

*5.3. Recommendations for Restoration Activities, Costs, and Funding*

The future versions of the platform will include specific information for selected areas according to the vegetation communities that were originally present. The prototype presented here incorporates generic text that is displayed when the mouse cursor is placed on the prioritized areas. In order to include restoration recommendations for each vegetation community, information needs to be compiled from diverse sources, including non-peer-reviewed publications such as technical documents and books, as well as scientific articles. Some of this information is already available for Chile in [23], but it needs to be updated and provided in more detail. Once this information is ready, estimated costs per hectare can be added for some of the activities. Finally, general information on funding opportunities to finance these activities will also be added.

**6. Conclusions**

This exercise shows that developing a platform to assess restoration priorities required expertise not only in remote sensing and GIS but also in ecological modeling and concepts and approaches from landscape ecology and ecological restoration. In order to integrate these different disciplines, you must develop multi-scale indicators. These indicators need

to faithfully represent landscape degradation at multiple geographic scales and inter-annual time periods, incorporating the relationship between degradation and the socio-economic situation in the study area (e.g., [33–35]). Likewise, the techniques applied need to allow the handling and analysis of large amounts of data in an autonomous manner with a high demand for computational processing. In spite of the multifaceted computational processing involved, the platform requires that it deliver the results of these complex analyses in simple, easy-to-interpret results with practical recommendations for restoration.

Currently, there are different software tools (e.g., Zonation, Marxan) that are capable of performing spatial prioritization to select sites and identify actions to develop management and/or conservation plans in areas of interest ([36,37]). However, these tools require downloading a computer program, collecting and generating all the required geospatial information, training in the use of the software, and running the program on computers with sufficient computing power to handle multiple layers. These requirements limit the use of these tools, especially in those cases where a large part of the necessary geospatial information needs to be collected, and even more so when the complexity of the problem and geographic extension of the territories under analysis require intensive use of human and financial resources and technical capabilities to implement the prioritization. In contrast, the solution proposed in this project will have all this geospatial information already loaded into the system, eliminating the requirement for users to make the effort to collect and generate the required indicators themselves. In addition, users will access the system through an intuitive and user-friendly graphical interface, which will be designed to minimize the need for GIS knowledge, thus maximizing the potential of the tool to be used by a wide and diverse target audience.

Another relevant point is that the proposed solution is based on free and open access geospatial information, and the data processing will be done through the "Google Earth Engine" system, which also corresponds to a free and open access system, and therefore the development of solutions does not require the use of technologies that are protected by patents. Furthermore, the platform does not require a specific cloud computing service, and is able to use any solution available in the market (e.g., Google Cloud Services, Amazon Web Services, or Microsoft Azure).

Furthermore, the improved versions of RePlant alpha will undoubtedly contribute to improving the experience, transparency, and efficiency of decision-making to prioritize the restoration of degraded ecosystems. In addition, the fact that it is an interactive platform where users can modify the relative importance of different indicators and dimensions will facilitate dialogue between decision-makers who may have conflicting opinions. In conjunction with these advantages, RePlant alpha can become a tool for scenario generation and discussion, as well as a base platform for developing other spatial prioritization objectives. In addition, this tool can contribute to the decision-making process for public policy instruments such as the National Strategy for Climate Change and Vegetation Resources and the National Landscape Restoration Plans, among others. At an international level, this type of tool can help in accomplishing targets and goals of commitments assumed by the countries for the restoration of degraded ecosystems, such as the Bonn Challenge, $20 \times 20$ Initiative, and the National Determined Contribution (NDC).

**Author Contributions:** Conceptualization, N.S.M. and I.C.F.; scripting and coding, N.S.M.; analysis, N.S.M., writing original draft, N.S.M., I.C.F., L.P.D. and W.A.P.-M.; writing—review & editing, N.S.M., I.C.F. and L.P.D.; figures, N.S.M., I.C.F. and W.A.P.-M. All authors have read and agreed to the published version of the manuscript.

**Funding:** The authors would like to thank the Center for Earth Observation (Hémera) for funding the platform's development.

**Data Availability Statement:** Not applicable.

**Acknowledgments:** The authors would like to thank Mario Valdivia for building the first coding draft and Giselle Muschett for her help revising this manuscript.

**Conflicts of Interest:** The authors declare no conflict of interest.

## Appendix A

**Table A1.** Prioritized area per administrative units in hectares, percentage of total prioritized area and percentage of commune area deemed of restoration efforts.

| Province | Commune | Area (ha) | Area (%) | % of Total Area |
|---|---|---|---|---|
| Los Andes (1%) | Calle larga | 4758 | 28.9 | 0.3 |
| | Los andes | 2684 | 16.3 | 0.2 |
| | Rinconada | 3096 | 18.8 | 0.2 |
| | San esteban | 5946 | 36.1 | 0.4 |
| | **Subtotal** | **16,484** | | |
| Marga Marga (1.8%) | Limache | 10,466 | 35.3 | 0.6 |
| | Olmué | 2559 | 8.6 | 0.2 |
| | Quilpué | 12,395 | 41.8 | 0.8 |
| | Villa Alemana | 4241 | 14.3 | 0.3 |
| | **Subtotal** | **29,661** | | |
| Petorca (2.2%) | Cabildo | 6921 | 19.4 | 0.4 |
| | La Ligua | 11,049 | 30.9 | 0.7 |
| | Papudo | 3600 | 10.1 | 0.2 |
| | Petorca | 7666 | 21.5 | 0.5 |
| | Zapallar | 6473 | 18.1 | 0.4 |
| | **Subtotal** | **35,709** | | |
| Quillota (1.6%) | Calera | 1790 | 7.1 | 0.1 |
| | Hijuelas | 3275 | 13.0 | 0.2 |
| | La Cruz | 3171 | 12.6 | 0.2 |
| | Nogales | 6572 | 26.1 | 0.4 |
| | Quillota | 10,403 | 41.3 | 0.6 |
| | **Subtotal** | **25,211** | | |
| San Antonio (4%) | Algarrobo | 13,900 | 21.6 | 0.9 |
| | Cartagena | 10,139 | 15.7 | 0.6 |
| | El quisco | 3075 | 4.8 | 0.2 |
| | El tabo | 7834 | 12.1 | 0.5 |
| | San Antonio | 21,297 | 33.0 | 1.3 |
| | Santo Domingo | 8239 | 12.8 | 0.5 |
| | **Subtotal** | **64,484** | | |
| San Felipe de Aconcagua (2.5%) | Catemu | 5738 | 14.4 | 0.4 |
| | Llaillay | 5720 | 14.3 | 0.4 |
| | Panquehue | 4996 | 12.5 | 0.3 |
| | Putaendo | 8237 | 20.7 | 0.5 |
| | San Felipe | 9270 | 23.2 | 0.6 |
| | Santa María | 5920 | 14.8 | 0.4 |
| | **Subtotal** | **39,881** | | |
| Valparaíso (5.5%) | Casablanca | 47,015 | 53.1 | 2.9 |
| | Concón | 2006 | 2.3 | 0.1 |
| | Puchuncaví | 6503 | 7.3 | 0.4 |
| | quintero | 10,704 | 12.1 | 0.7 |
| | Valparaíso | 17,452 | 19.7 | 1.1 |
| | Viña del mar | 4875 | 5.5 | 0.3 |
| | **Subtotal** | **88,555** | | |
| | **Total** | **299,985** | | |

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
