# Peer review of "RePlant Alfa: Integrating Google Earth Engine and R Coding to Support the Identification of Priority Areas for Ecological Restoration"

_land, doi:10.3390/land12020303_

Round 1

Reviewer 1 Report

RePlant alfa: integrating Google Earth Engine and R coding to support the identification of priority areas for ecological restoration

In this manuscript, the authors develop a semiautomatic geospatial platform to prioritize areas for restoration activities based on ecological, social and economic variables. This platform takes advantage of the potential to integrate R coding, Google Earth Engine cloud computing and GIS visualization services to generate an interactive geospatial decision-maker tool for restoration. For this purpose, they present a prototype version called "RePlant alpha" which was tested with data from the Central Zone of Chile. Their exercise proved that integrating R and GEE was feasible, and that the analysis, with at least six indicators and for a specific region was also feasible to implement even from a personal computer. Therefore, the use of a virtual machine in the cloud with a large number of indicators over large areas is both possible and practical.

After I have read this manuscript, I think the topic is interesting and the presentation and structure is pretty adequate. I can recommend it for publication after the next suggestions are considered:

1.       The authors start with “Land transformation and soil degradation are among the greatest threats to the resilience of ecosystems worldwide (Steffen et al, 2015). Thus, developing strategies to help restoring degraded ecosystems is an urgent task that will help keeping the planet in healthy ecological conditions to support biodiversity, tackle climate change, and promote human development (Abhilash, 2021)”. -> My suggestion at this point is to introduce a paragraph that contextualizes this generic aspect “land transformation and soil degradation”. So far, this presentation is too vague and imprecise. For example, both factors (land transformation and soil degradation) depend on factors like human pressure and activities over. One key aspect is for example, the urbanization process, which is showing heavy contrasts around the globe. For example, the fast-developed cities in southeastern Asia and Africa is leading to extreme risks due to the speed dynamics and the lack of governmental control. I could recommend you the study from Pentland et al. (2019) where the authors show the evolution of some representative global dynamics across the globe since 1960 in terms of increasing wealth and environmental costs –decoupling effect- (Globalization and the shifting centers of gravity of world's human dynamics: Implications for sustainability). The perspective of this work can help you to contextualize much better the risks and the existence of uneven spatial patterns at a global scale.

2.       Just a few lines ahead, the authors continue with “Spatial prioritizations like this could be addressed with the use of Multi-Criteria Spatial Decision Analyses, which integrate different geospatial variables through a system of interaction rules that combine them to generate information for decision-making (Mendoza & Martins, 2006; Malczewski 2006; Huang et al. 2011). Such an approach has been recently proposed to identify global priority areas for restoration (Strassburg et al, 2020), and has been also used for prioritizing areas for restoration at finer scales (e.g. Orsi & Geneletti 2010; Vettorazzi & Valente 2016; Fernández & Morales 2016).” I agree with the importance of bringing to the discussion the spatial data component (and spatial patterns). This is a key point of this research.

3.       Just after they argue: “Therefore, researchers could be tempted to reduce the participation of stakeholders in spatial prioritization methods to make the process simpler and with more generalizable outcomes, however decreasing the legitimacy of the results (Sharpe et al, 2021). This is also a key point that the authors bring (validity of the results and data extraction) and not so many studies do it. I would suggest them to emphasize this aspect by considering the particular aspects of the study area and the differences from any other areas. Some recent studies refer to this, and the importance of understanding the context, but also the scale of the data. In fact, the authors argue at some point “To integrate these different disciplines, you must develop multi-scale indicators. These indicators need to faithfully represent landscape degradation at multiple geographic scales and inter-annual time periods, incorporating the relationship between degradation and the socio-economic situation in the study area.” I can recommend you the study from Morales et al. (2022) “Scale, context, and heterogeneity: the complexity of the social space”, where the authors refer to this particular aspect. I would recommend you to extend a little bit this point by including some relevant references to recent studies.

4.       The authors argue later, “This may be particularly true for spatial analyses, as the tasks of data collection, processing and analysis are intense in terms of time and computational resources”. That is (referring to the availability and use of computational resources) also a key aspect that the authors consider in this manuscript. I agree with this.

5.       About the graphical part, I would suggest to make more clear the spatial scale in Figure 2 (it is difficult to realize it), but also including in the rest of the maps (Figure 3).

Author Response

Comments Reviewer 1 In this manuscript, the authors develop a semiautomatic geospatial platform to prioritize areas for restoration activities based on ecological, social and economic variables. This platform takes advantage of the potential to integrate R coding, Google Earth Engine cloud computing and GIS visualization services to generate an interactive geospatial decision-maker tool for restoration. For this purpose, they present a prototype version called "RePlant alpha" which was tested with data from the Central Zone of Chile. Their exercise proved that integrating R and GEE was feasible, and that the analysis, with at least six indicators and for a specific region was also feasible to implement even from a personal computer. Therefore, the use of a virtual machine in the cloud with a large number of indicators over large areas is both possible and practical. After I have read this manuscript, I think the topic is interesting and the presentation and structure is pretty adequate. I can recommend it for publication after the next suggestions are considered: 1. The authors start with “Land transformation and soil degradation are among the greatest threats to the resilience of ecosystems worldwide (Steffen et al, 2015). Thus, developing strategies to help restoring degraded ecosystems is an urgent task that will help keeping the planet in healthy ecological conditions to support biodiversity, tackle climate change, and promote human development (Abhilash, 2021)”. -> My suggestion at this point is to introduce a paragraph that contextualizes this generic aspect “land transformation and soil degradation”. So far, this presentation is too vague and imprecise. For example, both factors (land transformation and soil degradation) depend on factors like human pressure and activities over. One key aspect is for example, the urbanization process, which is showing heavy contrasts around the globe. For example, the fast-developed cities in southeastern Asia and Africa is leading to extreme risks due to the speed dynamics and the lack of governmental control. I could recommend you the study from Pentland et al. (2019) where the authors show the evolution of some representative global dynamics across the globe since 1960 in terms of increasing wealth and environmental costs –decoupling effect- (Globalization and the shifting centers of gravity of world's human dynamics: Implications for sustainability). The perspective of this work can help you to contextualize much better the risks and the existence of uneven spatial patterns at a global scale. A new paragraph was added to address this comment. 2. Just a few lines ahead, the authors continue with “Spatial prioritizations like this could be addressed with the use of Multi-Criteria Spatial Decision Analyses, which integrate different geospatial variables through a system of interaction rules that combine them to generate information for decision-making (Mendoza & Martins, 2006; Malczewski 2006; Huang et al. 2011). Such an approach has been recently proposed to identify global priority areas for restoration (Strassburg et al, 2020), and has been also used for prioritizing areas for restoration at finer scales (e.g. Orsi & Geneletti 2010; Vettorazzi & Valente 2016; Fernández & Morales 2016).” I agree with the importance of bringing to the discussion the spatial data component (and spatial patterns). This is a key point of this research. We are glad that the reviewer agrees with this important point 3. Just after they argue: “Therefore, researchers could be tempted to reduce the participation of stakeholders in spatial prioritization methods to make the process simpler and with more generalizable outcomes, however decreasing the legitimacy of the results (Sharpe et al, 2021).” This is also a key point that the authors bring (validity of the results and data extraction) and not so many studies do it. I would suggest them to emphasize this aspect by considering the particular aspects of the study area and the differences from any other areas. Some recent studies refer to this, and the importance of understanding the context, but also the scale of the data. In fact, the authors argue at some point “To integrate these different disciplines, you must develop multi-scale indicators. These indicators need to faithfully represent landscape degradation at multiple geographic scales and inter-annual time periods, incorporating the relationship between degradation and the socio-economic situation in the study area.” I can recommend you the study from Morales et al. (2022) “Scale, context, and heterogeneity: the complexity of the social space”, where the authors refer to this particular aspect. I would recommend you to extend a little bit this point by including some relevant references to recent studies. We addressed this comment in the new version adding a new paragraph in the introduction (lines 74-81). In addition, references were added to the conclusion section (line 332) as requested. 4. The authors argue later, “This may be particularly true for spatial analyses, as the tasks of data collection, processing and analysis are intense in terms of time and computational resources”. That is (referring to the availability and use of computational resources) also a key aspect that the authors consider in this manuscript. I agree with this. We are glad that the reviewer agrees with this important point 5. About the graphical part, I would suggest to make more clear the spatial scale in Figure 2 (it is difficult to realize it), but also including in the rest of the maps (Figure 3). The scale in figure 2 was changed to white. Scale was added to figure 3 as requested.

Reviewer 2 Report

The effort of identification of priority areas for ecological restoration is interesting and will bring a significant contribution in this field.

Besides this manuscript needs serious improvement.

It doesn’t clear from manuscript is proposed platform online (L78) or main part of this platform is offline (L99, Fig1)? If it is online – it is a good idea to describe pros and cons against Earth Engine. From the manuscript proposed platform looks similar to GEE. If it is online – part about cloud R computing should be added.

L274. Change R to Python3 looks strange. What is the sense to change computation language in future versions?

L155. What is the sense of using local computing of data that does not stored in GEE if you can simply load it as asset?

Alternatives of R packages should be described. Difference between GEE and RePlant should be specified too.

Reference list seems small as a review part and should be increased.

I wish that my comment would be helpful in improving the quality of this research.

Thank you.

Author Response

Reviewer 2 The effort of identification of priority areas for ecological restoration is interesting and will bring a significant contribution in this field. Besides this manuscript needs serious improvement. It doesn’t clear from manuscript is proposed platform online (L78) or main part of this platform is offline (L99, Fig1)? As described in lines 90-93, 95-101 and figure 1, some processes are performed locally and others in the cloud. The code is run locally by calling different cloud services. Newer versions will be completely hosted in the cloud, with the code running in a virtual machine in the cloud. In summary, the collection and preprocessing of satellite-based spatial data and the visual representation of results are both currently performed in the cloud. If it is online – it is a good idea to describe pros and cons against Earth Engine. From the manuscript proposed platform looks similar to GEE. If it is online – part about cloud R computing should be added. Despite that GEE can perform different spatial analysis and has a huge library of datasets does not have local small scale country specific databases at least for our study area. One of our main objectives was to evaluate is there was a need to virtualize the whole process. Currently, the code runs locally. However, there is no added complexity to run R in a cloud based virtual machine, is basically running an R session in a bigger mainframe. In that sense, is almost the same as running it locally only that you have access to high performance computing. L274. Change R to Python3 looks strange. What is the sense to change computation language in future versions? It is not a change in the original language code. If we choose in the future to develop our own GIS server similar to ArcGis online for instance, we need to code it. In general, the most used language in GIS is Python which is compatible with most GIS applications. ArcGis online itself is HTML and Java based. All the calculations and analysis will be carried out using R under the hood. L155. What is the sense of using local computing of data that does not stored in GEE if you can simply load it as asset? At the moment GEE is limited in the type of analysis that can perform. R has several packages and functions that are not available in GEE. However, if in the future GEE can have the same flexibility that R will be a game changer and we would think in changing everything to GEE. Alternatives of R packages should be described. Difference between GEE and RePlant should be specified too. We do not understand this comment, if the reviewer clarifies this will be helpful. Some of the R packages do not have alternatives. We cannot list differences between GEE and RePlant as GEE is part of Rplant coding. We think relevant to highlight at this point that our platform is not an alternative to GEE, but a platform that, by integrating GEE, R and a GIS online viewer, to provide a virtual setting for decision makers to evaluate the results of potential decision scenarios. Reference list seems small as a review part and should be increased. As a technical note the manuscript was focused on the application of different tools. That is one of the reasons why there are not so many references. Despite of that we added two paragraphs to the introduction, as per request of reviewer 1, which included a number of new references. Also, we added new references to the conclusions as requested by reviewer 1.

Round 2

Reviewer 2 Report

Authors had replied on all of the qustions and clarify all of the important parts of their paper. Due to the major improvements on first version of the manuscript, in my opinion it can be published in present form